# Current Status and Future Directions of Image-Guided Adaptive Brachytherapy for Locally Advanced Cervical Cancer

**DOI:** 10.3390/cancers16051031

**Published:** 2024-03-02

**Authors:** Nicholas Eustace, Jason Liu, Colton Ladbury, Andrew Tam, Scott Glaser, An Liu, Yi-Jen Chen

**Affiliations:** Department of Radiation Oncology, City of Hope National Medical Center, 1500 E Duarte Rd., Duarte, CA 91105, USA; neustace@coh.org (N.E.); jaliu@coh.org (J.L.); cladbury@coh.org (C.L.); atam@coh.org (A.T.); sglaser@coh.org (S.G.); aliu@coh.org (A.L.)

**Keywords:** brachytherapy, cervical cancer, image-guided brachytherapy, adaptive brachytherapy, magnetic resonance imaging, ultrasound-guided brachytherapy

## Abstract

**Simple Summary:**

Brachytherapy is a key component of radiation treatment in the curative treatment of locally advanced cervical cancer. To improve the delivery of brachytherapy, recent studies have explored the use of imaging, such as magnetic resonance imaging (MRI), in treatment planning. In this review, we reviewed the current evidence for image-guided brachytherapy and discussed future directions for research.

**Abstract:**

Purpose: The standard of care for patients with locally advanced cervical cancer is definitive chemoradiation followed by a brachytherapy boost. This review describes the current status and future directions of image-guided adaptive brachytherapy for locally advanced cervical cancer. Methods: A systematic search of the PubMed and Clinicaltrials.gov databases was performed, focusing on studies published within the last 10 years. The search queried “cervical cancer [AND] image-guided brachytherapy [OR] magnetic resonance imaging (MRI) [OR] adaptive brachytherapy”. Discussion: The retroEMBRACE and EMBRACE-I trials have established the use of MRI as the standard imaging modality for brachytherapy application and planning. Quantitative imaging and radiomics have the potential to improve outcomes, with three ongoing prospective studies examining the use of radiomics to further risk-stratify patients and personalize brachytherapy. Another active area of investigation includes utilizing the superior soft tissue contrast provided by MRI to increase the dose per fraction and decrease the number of fractions needed for brachytherapy, with several retrospective studies demonstrating the safety and feasibility of three-fraction courses. For developing countries with limited access to MRI, trans-rectal ultrasound (TRUS) appears to be an effective alternative, with several retrospective studies demonstrating improved target delineation with the use of TRUS in conjunction with CT guidance. Conclusions: Further investigation is needed to continue improving outcomes for patients with locally advanced cervical cancer treated with image-guided brachytherapy.

## 1. Introduction

Carcinoma of the cervix is the fourth most common cancer among women worldwide, with more than 99% of cases caused by the human papillomavirus (HPV). Worldwide, there are more than 500,000 cases of cervical cancer and approximately 250,000 deaths, with 80% of cases occurring in developing countries. In the United States, there are an average of 4000 deaths per year from cervical cancer. There are more than 130 different types of HPV, with types 16 and 18 responsible for 75% of cervical cancer cases. Cervical cancer deaths peak in middle-aged women aged 40–50 years; however, since the HPV vaccination was released in 2006, it has become a largely preventable disease as the vaccination has been highly effective at preventing the development of cervical cancer [1].

In the United States and Europe, brachytherapy boosts after definitive chemoradiation of 45–50.4 Gy is the standard of care for patients with locally advanced cervical cancer [2], which is classified by the International Federation of Gynecology and Obstetrics (FIGO) and includes stage IB2-IVA. Locally advanced cervical cancer is the most common cancer treated today using a combination of external beam radiation therapy and intracavitary brachytherapy [3]. Brachytherapy is a form of radiation therapy where a sealed radiation source is placed inside or next to the area requiring treatment and is sometimes called internal radiation. When the radiation source is placed inside a body cavity, such as through the cervix and into the uterus in the treatment of cervical cancer, this is called intracavitary brachytherapy, whereas interstitial brachytherapy is when the radiation source is inserted directly into tissue using needles or catheters. Intracavity brachytherapy is accomplished commonly using trans-abdominal ultrasound-guided tandem placement and vaginal packing, which has been shown to decrease rates of uterine perforation and reduce organ at risk (OAR) doses [4]. Brachytherapy takes advantage of the inverse square law, where the dose of radiation is inversely proportional to the square of the distance from the source [5]. Ensuring the close proximity of the radioactive source to the target allows for the delivery of very high doses of radiation to the tumor or at-risk areas, with low doses being delivered to adjacent normal critical structures such as the bladder and rectum [3]. The brachytherapy boost is typically delivered at a dose of 25–30 Gy in four to six fractions over two to three weeks after the completion of external beam radiation [6], and enables the high doses of radiation required to control cervical cancer (>80 Gy) to be delivered without too many side effects [5]. Overall, the effectiveness of brachytherapy relies on high-resolution image guidance to ensure that an adequate dose is being delivered to the tumor while minimizing the dose to nearby OARs including the bladder, rectum, sigmoid colon, and vagina.

In the past two decades, several advances in brachytherapy technique have led to improved cure rates and reduced toxicity, notably the implementation of magnetic resonance imaging (MRI) before and during brachytherapy application, as well as the adoption of adaptive, three-dimensional planning [7,8,9]. Today cure rates remain high with negative pelvic lymphadenopathy, gross tumor volume (GTV) dose covering 90% of the target (GTV D90) of >110 Gy, and a treatment duration of ≤56 days predictive of better outcomes [10]. These modern brachytherapy techniques allow for a more accurate treatment strategy that is necessary due to the high doses per fraction, internal organ motion, and changes in organ filling between fractions. Active areas for image-guided adaptive brachytherapy include (1) incorporating quantitative imaging and radiomics to identify poor responders to intensify treatment or conversely good responders for whom treatment may be de-escalated, (2) utilizing the superior soft tissue contrast provided by MRI to increase the dose per fraction and decrease the number of fractions needed for brachytherapy, and (3) utilizing trans-rectal ultrasound (TRUS) as an alternative to MRI in developing countries to improve target delineation. The purpose of this review is to provide perspective on the history, current status, and future directions of image-guided adaptive brachytherapy for cervical cancer.

## 2. Materials and Methods

A systematic search of the PubMed and Clinicaltrials.gov database was performed, focusing on studies published within the last 10 years (2014–2024). The search queried “(cervical cancer) AND ((image-guided brachytherapy) OR (magnetic resonance imaging) OR (adaptive brachytherapy))”. This query on PubMed resulted in 2395 results, which were further limited with the inclusion of prospective and retrospective studies of at least 50 patients and the exclusion of preclinical studies, preprints, abstracts, book chapters, documents, and reviews. This narrowed the result to 63 articles, all of which were individually screened for relevance. Furthermore, we included any papers discovered outside the search criteria through references for a total of 31 articles undergoing in-depth analysis in this subsequent review.

## 3. Discussion

### 3.1. Historical Overview

Brachytherapy was originally performed using two-dimensional images and the dose was prescribed to a reference point located 2 cm superior to the lateral vaginal fornix and 2 cm lateral to the cervical canal called Point A [11]. Over the past two decades, advancements in imaging technologies have enabled new methods of guidance for brachytherapy. The initial steps were made with CT-guided brachytherapy [12] and later with MRI-guided brachytherapy [8,9,13,14]. Over time, two-dimensional, point-A-based planning was replaced by three-dimensional, adaptive planning. The Groupe Européen de Curiethérapie—European Society for Radiotherapy & Oncology (GEC-ESTRO)—published recommendations on target definition and dosimetric parameters to standardize the reporting and delivery of brachytherapy [15,16]. They proposed three volumes:

A low-risk clinical target volume (LR-CTV), comprising all potentially involved tissues, corresponding to external beam radiation targets.

A high-risk CTV (HR-CTV), comprising the whole cervix and residual disease at the time of brachytherapy.

An intermediate-risk CTV (IR-CTV), comprising the HR-CTV with directional margins, and at least the initial volume at diagnosis.

Several series comparing three-dimensional planning versus (vs.) two-dimensional planning have been published [17,18,19]. The first two were retrospective studies that found a significant improvement in overall survival (OS) and toxicity with the use of three-dimensional planning vs. two-dimensional planning. However, these two studies had notably higher rates of concurrent chemotherapy use in the three-dimensional planning arm, which makes it difficult to attribute the improvement in outcomes to the treatment planning technique alone. The STIC trial [19] overcame these limitations by comparing two nonrandomized cohorts formed prospectively and in parallel. Their results showed that three-dimensional planning had significantly better local control (LC) (78.5% vs. 73.9% at two years) and Common Terminology Criteria for Adverse Events (CTCAE) grade 3–4 toxicity (2.6% vs. 22.7%) compared to two-dimensional planning.

Shortly after the publication of its recommendations, GEC-ESTRO launched two studies: EMBRACE-I (an intErnational study on MRI-guided BRAchytherapy in locally Advanced Cervical cancer) [20] and Retro-EMBRACE [21]. EMBRACE-I was a prospective, observational, multicenter cohort study published in 2021 that included 1416 patients with cervical cancer treated with definitive chemoradiation followed by MRI-guided brachytherapy using T2-weighted spin echo sequences [20]. After a median follow-up of 51 months, LC was 92% at five years, OS was 74% at five years, and the incidence of grade 3–5 toxicities was 7% for genitourinary events, 9% for gastrointestinal events, and 6% for vaginal events. Retro-EMBRACE was a retrospective study published in 2016 that included 731 patients with cervical cancer treated with definitive chemoradiation followed by MRI-guided (81%) or CT-guided (19%) brachytherapy [21]. After a median follow-up of 43 months, LC was 89% at five years, OS was 73% at five years, and the incidence of grade 3–5 toxicities was 5% for genitourinary events, 7% for gastrointestinal events, and 5% for vaginal events. These results confirm the mono-institutional results and seem superior to results from prior studies, especially in advanced lesions. Based on these impressive results, MRI-guided brachytherapy using T2-weighted images (T2WI) has been established as the standard of care approach for patients with locally advanced cervical cancer [2].

### 3.2. Ongoing Areas of Investigation

Building upon the success of the EMBRACE-I trial [20], several studies have aimed to further personalize brachytherapy for patients with cervical cancer using radiomics. Radiomics is a quantitative approach to medical imaging, which aims at enhancing the existing data available to clinicians using advanced mathematical analysis [22]. By extracting the spatial distribution of signal intensities and pixel interrelationships, radiomics can quantify textural information for analysis via machine learning which can automate target and OAR delineation or anticipate patterns of failure. The typical workflow of a radiomics analysis is as follows: (1) image acquisition and segmentation, (2) image processing, (3) feature extraction, (4) feature selection/dimension reduction, and (5) model training and classification [23].

Several retrospective studies have been published showing the potential for radiomic features to predict outcomes for patients with cervical cancer [24,25,26,27,28,29,30]. For example, a preoperative retrospective analysis (N = 104) of MRI texture determined on T2WI + diffusion-weighted imaging (DWI) was found to perform well at predicting lymph node metastasis better than morphologic criteria of lymph node status determined by MRI in patients with cervical non-SCC. T2WI + DWI was also sufficient on its own to be good at predicting metastasis and accurate discernment was not significantly improved with the inclusion of contrast-enhanced T1WI [30].

Due to the retrospective nature of these studies, there are several limitations. The reproducibility of these studies is poor due to a lack of standardization, insufficient reporting, or limited open-source code and data. Furthermore, radiomic feature values are influenced by variabilities in scanners, settings, and patient geometry, which impact the levels of noise and the presence of artifacts in an image. Because these studies are based on retrospectively collected data with a low level of evidence, they should be seen as “proof-of-concept,” whereas prospective studies are required to confirm the value of radiomics. Currently, there are three ongoing prospective studies examining the utility of quantitative imaging and radiomics to predict response in patients with cervical cancer. A summary of these studies is shown in Table 1.

The first active study, IQ-EMBRACE [31], is an observational, nonrandomized sub-study under the EMBRACE-II protocol in which patients who were included in the prior EMBRACE-II study are eligible to enroll. The MRI evaluation in this study will be more comprehensive than current clinical practice in that the MRIs will include T1, T2, DWI (quantifies diffusion of water molecules within a tissue), and DCE (quantifies perfusion within a tissue) sequences. At the time of brachytherapy, the treatment plan’s MRI will additionally include DWI and quantitative T2 (qT2). The study’s primary objective is to assess the sensitivity and specificity of dynamic contrast-enhanced (DCE) MRI to identify patients at increased risk of disease recurrence after radio-chemotherapy of cervical cancer. The secondary objectives include using radiomics to identify patients at increased risk of disease recurrence after radio-chemotherapy, to correlate MR imaging parameters with biomarkers based on pathology, and ultimately to evaluate the implementation of quantitative imaging in a multicenter setting.

DCE MRI is utilized to serve as a surrogate marker for hypoxic regions that are poorly perfused with the rationale that hypoxic tumor cells are more radiation resistant and have increased metastatic potential [31]. Hypoxia has long been shown to be a major cause of radiation resistance in various tumor sites and is a factor associated with poor outcomes for patients with cervical cancer [34]. DCE MRI is the most investigated functional imaging technique for locally advanced cervical cancer and prior studies have found an association between low DCE signal [35,36,37] and treatment failure. However, a significant diversity of methodology exists between these studies, and not all papers show a predictive effect [31]. DWI MRI has also been investigated in cervical cancer and a high apparent diffusion coefficient (ADC) (magnitude of diffusion within a tissue) is predictive of poor treatment response [38,39,40,41,42]. Overall, IQ-EMBRACE is the largest ongoing prospective study, with a target accrual of 320 patients, and has the potential to benefit patients through improvements in risk stratification.

The second study, EMPIRIC [32], from the University of Manchester, is an exploratory study using multiparametric MRI (mpMRI) to develop a method to predict which cervical cancer patients are less likely to respond well to chemoradiation, and then intervene with increased treatment intensity. Patients will be scanned with three different types of MRI sequences (DWI, DCE, and blood-oxygen-level-dependent [BOLD] imaging) at three different time points during chemoradiation to measure the degree and change in hypoxia throughout treatment. If high levels of hypoxia are identified using mpMRI, a prognostic imaging biomarker model can be developed to predict treatment outcomes of patients with locally advanced cervical cancer and potentially allow for dose escalation to hypoxic areas. The pre-planned radiomic features to be analyzed in this study include the extravascular extracellular space volume fraction V_e_ (%), transfer constant K_trans_ (1/min), rate constant k_ep_ (1/min), blood volume fraction V_p_ (%), apparent diffusion coefficient (ADC) (um^2^/s), BOLD-based reversible transverse relaxation rate R2 (1/s), relaxation times (1/s), and second-order Haralick textural radiomic features. This single-institution study aims to accrue 40 patients and is recruiting at the time of this publication.

The third study, RaPiCCa [33], from the University of Copenhagen, is a pilot study that aims to evaluate the potential and feasibility of hybrid PET/mpMRI functional imaging for external beam radiation and brachytherapy planning for cervical cancer. The radiotracer used in this study is “RGD” (68Ga-NODAGA- E[c(RGDyK)]2), which identifies areas of angiogenesis within the tumor. Hybrid PET/mpMRI has the potential to improve target volume delineation for radiation treatment planning compared to MRI alone given the additional information on tumor heterogeneity, perfusion characteristics, and hypoxia provided by PET, DW-MRI, and DCE-MRI. Another endpoint of interest for this study is to better understand the underlying pathology and changes in angiogenesis within the tumor during treatment using the RGD radiotracer. Investigators expect increased angiogenesis of individual tumors to correlate with worsened LC and poorer prognosis, which may suggest a role for dose escalation in these tumors. This single-institution study aims to accrue 25 patients.

Ultimately, if these quantitative MRI techniques are found to be predictive of treatment response in cervical cancer, it will improve our understanding of treatment failures and enable future studies into treatment escalation for patients at a high risk of failure or de-escalation for low-risk patients, leading to more effective therapies.

### 3.3. Utilizing MRI to Decrease the Number of Fractions for Brachytherapy

With the improved soft tissue visualization from MRI for brachytherapy planning, several studies have examined whether the dose per fraction may be increased and the total number of fractions may be decreased for brachytherapy. Typically, brachytherapy for cervical cancer is performed over four to five fractions in two to three weeks. Efforts to reduce the number of fractions for brachytherapy stem from the fact that (1) it is inconvenient and oftentimes uncomfortable for patients to undergo each fraction of brachytherapy, (2) it is very time-consuming and labor-intensive, requiring a multidisciplinary team of anesthetists, oncologists, radiographers, nurses, and physicists for each fraction of brachytherapy, and (3) it allows room for treatment delays if needed while keeping the overall treatment time from the start of chemoradiation to the end of brachytherapy <8 weeks, as recommended by national guidelines [43].

Currently, there have been a few studies examining two- or three-fraction regimens for brachytherapy [44,45,46,47]. These studies have generally been performed in resource-limited countries as a way to preserve resources and decrease the treatment time. However, they often are performed with two-dimensional planning without MRI guidance, and as a result, have shown much higher rates of acute toxicity compared to modern series. This has led to some reluctance to adopt two- or three-fraction regimens in the United States and Europe. However, with the use of MRI guidance and three-dimensional, adaptive planning, it is believed that two- or three-fraction regimens may result in comparable outcomes with the standard four- or five-fraction regimens. A summary of these studies is shown in Table 2.

The first by Guyader et al. [48] was an observational, retrospective, single-institution study in France that included 191 patients with locally advanced cervical cancer treated between 2007 and 2018. Patients were treated with four different schemes using twice-a-day (BID) MRI-guided brachytherapy: 7 Gy/1 fraction (fx) + 3.5 Gy/4 fx (group 1); 7 Gy/1 fx + 4.5 Gy/4 fx (group 2); 7 Gy/3 fx (group 3) and 8 Gy/3 fx (group 4). After a median follow-up of 57 months, the five-year progression-free survival (PFS) was 61% (95% CI 57–64%) and the five-year OS was 75% (95% CI 69–78%), with no significant difference between the groups. The five-year PFS was 58% in patients treated with a five-fraction brachytherapy course (groups 1–2) vs. 64% in patients treated with a three-fraction brachytherapy course (groups 3–4). The five-year OS was 76% in patients treated with a five-fraction brachytherapy course (groups 1–2) vs. 75% in patients treated with a three-fraction brachytherapy course (groups 3–4). The rate of grade 2 or higher urinary and gastrointestinal toxicities was similar between all groups, although slightly higher in groups 2 and 4 (dose escalation arms). The authors concluded that BID brachytherapy delivered in three fractions is feasible, with similar oncological outcomes as brachytherapy delivered in five fractions.

The second study by Scott et al. [49] was a dosimetric, retrospective, single-institution study from the Princess Margaret Cancer Center that included 224 patients with locally advanced cervical cancer treated between 2016 and 2021. Most patients in this study had squamous cell carcinoma and T2b disease and were treated with an intracavitary applicator plus interstitial needles (96%). The patients were treated with 24 Gy in three fractions (n = 133) vs. 28 Gy in four fractions (n = 91) using MRI guidance. The authors found that patients treated with three-fraction brachytherapy had comparable target and OAR doses compared to those treated with four-fraction brachytherapy and there were no significant differences in the proportion of patients able to meet the EMBRACE II OAR dose constraints, with the exception that fewer patients treated with 28 Gy in four fractions were able to meet D_2cm_^3^ < 65 Gy at 73% vs. 85% (*p* = 0.027), respectively, or the International Commission on Radiation Units and Measurements (ICRU) rectovaginal point < 65 Gy with 65% vs. 84% (*p* = 0.005), respectively. The mean IR-CTV D98% was 64.5 Gy vs. 65.5 Gy, the mean rectum D2cc was 59.2 Gy vs. 61.7 Gy, and the mean bladder D2cc was 77.9 Gy vs. 81.3 Gy between the three-fraction and the four-fraction brachytherapy arms, respectively. The authors concluded that a less resource-intense fractionation schedule of 24 Gy in three fractions can be considered a safe alternative to 28 Gy in four fractions for brachytherapy.

The third study by Williamson et al. [50] was a retrospective, single-institution study from the University of California, San Diego, that included 150 patients with locally advanced cervical cancer treated between 2015 and 2020. The patients were treated with 24 Gy in three fractions (n = 32) vs. 28–30 Gy in four to five fractions (n = 118) using MRI guidance. The two-year local failure rate was similar between the two groups, which was 3.6% for the patients treated with three-fraction brachytherapy vs. 7.5% for the patients treated with four- or five-fraction brachytherapy (HR 0.43, 95% CI 0.05–3.43, *p* = 0.43). The toxicities were also similar between the two groups, with a rate of grade 3 or higher toxicity of 6.3% in the patients treated with three-fraction brachytherapy vs. 5.9% in the patients treated with four- or five-fraction brachytherapy (*p* = 1.00). There was no difference in the hospitalizations between the two groups within two years (*p* = 0.66) and no treatment-related deaths. The authors concluded that three-fraction brachytherapy has excellent LC and similar toxicity to four- or five-fraction brachytherapy. Further prospective studies are warranted to establish the safety and efficacy of three-fraction courses before widespread adoption.

### 3.4. Utilizing Trans-Rectal Ultrasound as an Alternative to MRI in Developing Countries

The availability of MRI remains limited, especially in developing countries, where the incidence and mortality from cervical cancer remain higher. A natural alternative to MRI in these situations would be using CT guidance for brachytherapy planning. However, data have shown that delineating CT leads to an overestimation of the CTVs compared to MRI [51]. This pitfall can be partly overcome by trimming the volumes according to clinical and initial imaging findings [52], although there are no standard recommendations or guidelines to follow.

A promising and inexpensive imaging modality that can be used to improve target delineation in conjunction with CT is trans-rectal ultrasound (TRUS). A study by Schmid et al. [53] compared maximum HR-CTV dimensions defined using MRI, CT, and TRUS and found that TRUS was more effective than CT at finding the maximum HR-CTV dimension. Differences observed between TRUS and MRI could be attributed to image slice orientation and the deformation of tissues by the probe, but, overall, there was good concordance between TRUS and MRI. Another study by Nesvacil et al. [54] compared TRUS/CT-based planning to MRI-based planning and found that there was high fidelity between dose values reported for TRUS/CT and MRI-only reference contours. The authors concluded that TRUS/CT-based treatment planning is feasible and may be clinically comparable to MRI-based treatment planning. Further development to solve challenges with applicator definition in the TRUS volume is required before this is adopted into routine practice in developing countries.

Currently, there are no ongoing prospective studies in the United States or Europe examining the utility of TRUS for cervical cancer brachytherapy planning, since MRI is commonly used in these countries. Certain centers within Japan and India (such as Kobe University Hospital and Tata Memorial Hospital) routinely utilize TRUS in their workflow and have published their experiences [55,56], which could serve as frameworks to standardize the use of TRUS for brachytherapy application and planning in other countries.

## 4. Conclusions

Several advances in image-guided adaptive brachytherapy over the past two decades have significantly improved outcomes for patients with locally advanced cervical cancer. The use of trans-abdominal ultrasound for tandem placement and vaginal packing has made the procedure itself safer by decreasing the rates of uterine perforation, and should be routinely and widely utilized [4]. Additionally, the retroEMBRACE and EMBRACE-I trials have established the use of MRI T2WI as the “gold standard” imaging modality for brachytherapy planning, and retrospective studies have found that using radiomic features from T2WI + DWI is useful for predicting metastasis and merit prospective validation [20,21]. Several exciting prospective studies are currently underway examining the utility of quantitative imaging and radiomics to further risk-stratify patients and personalize brachytherapy, including the IQ-EMBRACE [31], EMPIRIC [32], and RaPiCCa [33] trials. Utilizing the superior soft tissue contrast provided by MRI to increase the dose per fraction and decrease the number of fractions needed for brachytherapy is another area of active investigation, with three retrospective studies [48,49,50] demonstrating the safety and feasibility of three-fraction courses. For developing countries with limited access to MRI, TRUS appears to be an effective alternative to MRI, with several studies demonstrating improved target delineation with the use of TRUS in conjunction with CT guidance [53,54]; however, the implementation of these new risk stratification techniques using radiometric features are limited by the locally available technology and will require expanding access to MRI and radiomics protocols worldwide. The most promising areas for improvements in image-guided brachytherapy lie in the successful development and implementation of radiomics techniques that will be further strengthened by ongoing advancements in machine learning to one day better anticipate areas of failure or toxicity to further personalize therapeutic targeting and dosing for the individual in this curable disease. Overall, the future remains bright for advancing outcomes in patients with locally advanced cervical cancer across the globe using both established and novel techniques for image guidance brachytherapy. 

## Figures and Tables

**Table 1 cancers-16-01031-t001:** Ongoing prospective studies examining the utility of quantitative imaging and radiomics.

Study	Patients	Imaging	Outcomes
IQ-EMBRACE [31]	N = 320 Cervical cancer patients eligible for EMBRACE-II	MRI T1, T2, DWI, DCE	Disease control, radiomics
EMPIRIC [32]	N = 40 FIGO stage IB-IVA cervical cancer	MRI DWI, DCE, BOLD	PFS, OS, radiomics
RaPiCCa [33]	N = 25 Cervical cancer with tumor >1 cm	MRI DWI, DCE “RGD” (68Ga-NODAGA-E[c(RGDyK)]2) PET	Feasibility of PET/MRI in workflow, radiomics of novel radiotracer, change in PTV between conventional MRI and PET/MRI

MRI = magnetic resonance imaging, PET = positron emission tomography, DWI = diffusion weighted imaging, DCE = dynamic contrast-enhanced, BOLD = blood-oxygen-level-dependent, PFS = progression-free survival, OS = overall survival.

**Table 2 cancers-16-01031-t002:** Retrospective studies examining the safety of brachytherapy delivered in three to four fractions.

Study	Patients	Dose	Outcomes
Guyader et al. [48]	N = 191 Cervical cancer	7 Gy + 4 × 3.5 Gy (group 1) vs. 7 Gy + 4 × 4.5 Gy (group 2) vs. 3 × 7 Gy (group 3) vs. 3 × 8 Gy (group 4)	Five-year PFS and OS similar between arms Toxicity similar between arms
Scott et al. [49]	N = 224 FIGO stage IB-IVA cervical cancer	3 × 8 Gy (n = 133) vs. 4 × 7 Gy (n = 91)	Target coverage and dose to OARs similar between arms
Williamson et al. [50]	N = 150 Cervical cancer	3 × 8 Gy (n = 32) vs. 4 × 7 Gy or 5 × 6 Gy (n = 118)	LC, toxicity, and hospitalizations similar between arms

PFS = progression-free survival, OS = overall survival, LC = local control.

## Data Availability

Supporting data can be found in the literature cited in this article.

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
