# Peer review of "Current Status and Future Directions of Image-Guided Adaptive Brachytherapy for Locally Advanced Cervical Cancer"

_cancers, 2024, doi:10.3390/cancers16051031_

Round 1

Reviewer 1 Report

Comments and Suggestions for Authors

The manuscript provides a comprehensive overview of the history, current status, and future directions of image-guided adaptive brachytherapy for locally advanced cervical cancer. It explores the efforts to decrease the number of fractions for brachytherapy by MRI guidance precision brachytherapy, ongoing areas of investigation, including the incorporation of quantitative imaging and radiomics to personalize brachytherapy. The manuscript also addresses the use of transrectal ultrasound (TRUS) as an alternative to MRI in developing countries with limited access to advanced imaging. TRUS, in conjunction with CT, has shown promise in improving target delineation for brachytherapy planning.

There is only some content that needs to be corrected including the reference:

1. In discussion section 3.3 (page 6, lines 243-244), the patient number and dose per fraction from studies (references 43) were discrepancies with the content of Table 2. Please correct it.

2. In discussion section 3.2 (page 3, lines 115-116), the sentence seems to be incomplete. Please confirm it.

Author Response

  1. The table dose and the patient number have been corrected.
  2. The sentence has been removed, and the paragraph updated in the new version of the manuscript.

Reviewer 2 Report

Comments and Suggestions for Authors

This is a very short review paper. 

I cannot see any research questions nor any explanation of how authors get those papers (like what databases are used).

Exclusion or inclusion criteria used?

Any limitations?

Why did authors use only the PubMed and Clinicaltrials.gov databases and not the Scopus?

Comments on the Quality of English Language

.

Author Response

The manuscript was updated and the total length increased, with additional introduction, discussion, and detail including inclusion/exclusion criteria, use of PubMed and Clinicaltrials.gov databases, and limitations. We did not see major differences in clinical trials that resulted in papers on Scopus vs the included databases and is why it was not mentioned.

Reviewer 3 Report

Comments and Suggestions for Authors

 The topic is exciting and a current area of investigation. More data is needed to evaluate the use of image guided brachy (and radiomics) for cervix cancer

The methodology is well described and appropriate.

The results are reflective of the findings, extensively described and easy to understand

Conclusions are appropriate and appropriately discuss limitations of the study citing need for further investigation

Author Response

Thank you! Additional introduction information, citations, and discussion are included in the revised manuscript per other reviewer requests.

Reviewer 4 Report

Comments and Suggestions for Authors

The authors presented the paper "Current Status and Future Directions of Image-Guided Adaptive Brachytherapy for Locally Advanced Cervical Cancer"

1) I recommend inserting some more keywords such as cervical cancer, image-guided brachytherapy, magnetic resonance imaging, adaptive brachytherapy

2) The reference list for the Introduction section should be improved to show the progress of the area. Only five references and such little Introduction is not enough.

3) Please, clearly mention what type of MRI is provided before brachytherapy in the paper text. There are a number of MRI methods: T1 or T2 images, MRI program parameters, what contrast agents, etc.

4) Some limitations and future outlook are highly recommended for the Conclusion section.

Minor

line 116 spaces

Comments on the Quality of English Language

Minor editing of English language required

Author Response

1) I recommend inserting some more keywords such as cervical cancer, image-guided brachytherapy, magnetic resonance imaging, and adaptive brachytherapy.
Keywords have been updated to include these words as well as "ultrasound-guided brachytherapy". 

2) The reference list for the Introduction section should be improved to show the progress of the area. Only five references and such little Introduction is not enough.
The introduction section has been expanded with additional detail and citations.  

3) Please, clearly mention what type of MRI is provided before brachytherapy in the paper text. There are a number of MRI methods: T1 or T2 images, MRI program parameters, what contrast agents, etc.

The manuscript is updated with specifics of MRI sequences that are used in specific studies and are useful.

4) Some limitations and future outlook are highly recommended for the Conclusion section.

The conclusion section has been expanded with additional discussion on current limitations and future outlook. 

2/23/2024: I have made requested edits to using MDPI styles for endnote and, modifying the grammar of the manuscript.

Round 2

Reviewer 2 Report

Comments and Suggestions for Authors

.

Author Response

No comments or suggestions were left by the reviewer, only a period. Nothing I am able to address.